# Three-Dimensional Flower-like MoS_2_ Nanosheets Grown on Graphite as High-Performance Anode Materials for Fast-Charging Lithium-Ion Batteries

**DOI:** 10.3390/ma16114016

**Published:** 2023-05-27

**Authors:** Yeong A. Lee, Kyu Yeon Jang, Jaeseop Yoo, Kanghoon Yim, Wonzee Jung, Kyu-Nam Jung, Chung-Yul Yoo, Younghyun Cho, Jinhong Lee, Myung Hyun Ryu, Hyeyoung Shin, Kyubock Lee, Hana Yoon

**Affiliations:** 1Korea Institute of Energy Research (KIER), Daejeon 34129, Republic of Korea; yeonga1902@kier.re.kr (Y.A.L.); kyuyeonjang@gmail.com (K.Y.J.); mitamire@kier.re.kr (K.-N.J.); jinhong02@kier.re.kr (J.L.); nicengood@kier.re.kr (M.H.R.); 2Graduate School of Energy Science and Technology (GEST), Chungnam National University, Daejeon 34134, Republic of Korea; yjs567@naver.com; 3Department of Advanced Energy Technologies and System Engineering, Korea University of Science and Technology (UST), Daejeon 34113, Republic of Korea; 4Computational Science and Engineering Laboratory, Korea Institute of Energy Research (KIER), Daejeon 34129, Republic of Korea; khyim@kier.re.kr (K.Y.); kyjung1020@kier.re.kr (W.J.); 5Department of Physics, Chungnam National University, Daejeon 34134, Republic of Korea; 6Department of Chemistry, Mokpo National University, Muan-gun 58554, Republic of Korea; chungyulyoo@mokpo.ac.kr; 7Department of Energy Systems, Soonchunhyang University, Asan 31538, Republic of Korea; yhcho@sch.ac.kr

**Keywords:** graphite, molybdenum disulfide, fast charging, high rate capability, hydrothermal synthesis, lithium-ion battery, anode materials

## Abstract

The demand for fast-charging lithium-ion batteries (LIBs) with long cycle life is growing rapidly due to the increasing use of electric vehicles (EVs) and energy storage systems (ESSs). Meeting this demand requires the development of advanced anode materials with improved rate capabilities and cycling stability. Graphite is a widely used anode material for LIBs due to its stable cycling performance and high reversibility. However, the sluggish kinetics and lithium plating on the graphite anode during high-rate charging conditions hinder the development of fast-charging LIBs. In this work, we report on a facile hydrothermal method to achieve three-dimensional (3D) flower-like MoS_2_ nanosheets grown on the surface of graphite as anode materials with high capacity and high power for LIBs. The composite of artificial graphite decorated with varying amounts of MoS_2_ nanosheets, denoted as MoS_2_@AG composites, deliver excellent rate performance and cycling stability. The 20−MoS_2_@AG composite exhibits high reversible cycle stability (~463 mAh g^−1^ at 200 mA g^−1^ after 100 cycles), excellent rate capability, and a stable cycle life at the high current density of 1200 mA g^−1^ over 300 cycles. We demonstrate that the MoS_2_-nanosheets-decorated graphite composites synthesized via a simple method have significant potential for the development of fast-charging LIBs with improved rate capabilities and interfacial kinetics.

## 1. Introduction

Among various energy storage technologies, lithium-ion batteries (LIBs) have been widely investigated as power sources for portable electronics and electric vehicles (EVs) due to their high energy density and long lifespan [1,2,3,4]. The rapid expansion of the global EV and energy storage systems (ESSs) market has led to significant demand for fast-charging battery technology that can support the high power and long cycle life requirements of these applications. Graphite, a commercial anode material used in LIBs, is still considered the most promising material because of its excellent cycle reversibility, stable cycle life, and superior electronic conductivity. Despite these advantages, due to its low theoretical capacity (~372 mAh g^−1^) and limited rate capability under fast charging conditions, graphite cannot meet the growing performance requirements of LIBs. The slow kinetics of lithium intercalation at the graphite–electrolyte interface during rapid charging conditions can lead to an undesirable anode voltage drop below 0 V vs. Li/Li^+^, resulting in the formation of lithium plating on the graphite surface. This phenomenon can cause capacity decay, dendrite growth, short circuits, and even serious safety issues [5,6,7,8]. To overcome this challenge, various approaches have been proposed to develop high-power lithium-ion battery (LIB) anodes with improved interfacial kinetics [9,10,11,12,13,14,15]. Among these approaches, surface modification using functional materials has emerged as an effective strategy to enhance the rate capability and cycle stability of graphite anodes [7,8,12,13,14,15].

Transition metal sulfides (TMDs) have been extensively investigated in various energy storage applications, such as batteries, supercapacitors, and electrocatalysts [16,17,18]. Among various TMDs, molybdenum disulfide (MoS_2_), a typical two-dimensional (2D) layered material, has emerged as a promising anode material for LIBs due to its high theoretical capacity (~670 mAh g^−1^), which is much higher than that of graphite [19]. Furthermore, the interlayer spacing of MoS_2_ (~0.62 nm) is larger than that of graphite (0.34 nm), providing ample space for fast Li^+^ diffusion paths [20,21]. However, MoS_2_ suffers from rapid capacity decay and inferior rate capability due to its low intrinsic electronic conductivity, large volume variation during cycling, and the production of polysulfide dissolution during the charge and discharge process [22,23,24,25,26].

Herein, we report a facile and efficient approach to synthesizing a 3D hierarchical MoS_2_/artificial graphite (MoS_2_@AG) composite for use as an anode material in LIBs. The composite was prepared using a hydrothermal reaction to decorate 3D hierarchically aligned flower-like MoS_2_ nanosheets directly on the surface of graphite. The resulting composite exhibits improved specific capacity and rate capability compared to commercial graphite anodes. The graphite matrix provides a conductive pathway for fast electron transfer through the electrode, while the 3D hierarchically aligned MoS_2_ nanosheets form a stable interface with the graphite, effectively preventing structural degradation and providing excellent electron transport. Moreover, the 3D hierarchical morphology of the MoS_2_ nanosheets enhances the electrode/electrolyte contact area, facilitating charge transfer kinetics. As a result, the MoS_2_-decorated graphite composite demonstrates exceptional rate capability, achieving charging times of less than 20 min for approximately 84% of its capacity. The composite also exhibits superior cycle stability under fast charging conditions compared to commercial graphite anodes. These results highlight the potential of the MoS_2_/graphite composite as a promising candidate for high-energy and high-power LIBs.

## 2. Materials and Methods

### 2.1. Preparation of MoS_2_- and MoS_2_-Decorated Graphite Composites

The MoS_2_-decorated graphite composite was synthesized via a facile hydrothermal reaction. To prepare the precursor solution, hexaammonium molybdate tetrahydrate ((NH_4_)_6_Mo_7_O_24_⋅4H_2_O) and thiourea (NH_2_CSNH_2_), both purchased from Sigma Aldrich, were dissolved in a molar ratio of 1:14 in 50 mL of water dispersion solution containing artificial graphite (Hitachi Chemical Co., Japan) at a concentration of 0.0342 g mL^−1^. The mixture was vigorously stirred for 1 h. Subsequently, the mixture was transferred into a Teflon-lined stainless autoclave and reacted at 220 °C for 16 h. After cooling naturally, the black precipitates were collected by centrifugation, washed with distilled water and ethanol, and dried at 80 °C for 24 h. The amount of thiourea and (NH_4_)_6_Mo_7_O_24_⋅4H_2_O added to the graphite solution varied to achieve mass ratios of 2.5:100, 5:100, 10:100, 20:100, or 30:100 and the corresponding products were denoted as 2.5−MoS_2_@AG, 5−MoS_2_@AG, 10−MoS_2_@AG, 20−MoS_2_@AG, or 30−MoS_2_@AG. The preparation process for the MoS_2_-decorated graphite composite is illustrated in Figure 1a. The synthesis procedure for a pristine flower-like MoS_2_ was similar to that of MoS_2_-decorated graphite composite but without the presence of graphite.

### 2.2. Characterizations

Powder X-ray diffraction (XRD) was measured on an X-ray diffractometer (Rigaku SmartLab High Resolution, Tokyo, Japan) using Cu-Kα radiation (λ = 1.5406 Å). Raman spectroscopy measurements were taken using a Raman spectrometer (HORIBA, LabRAM HR Evolution, Lyon, France) with an excitation wavelength of 514 nm. X-ray photoelectron microscopy (XPS) analysis was conducted using a Thermo VG Scientific Sigma Probe instrument with a micro-focused monochromatized Al Ka X-ray source (1486.6 eV). Field-emission scanning electron microscopy (FE-SEM) images were acquired using a FEI NovaNano SEM 450, and transmission electron microscopy (TEM) with energy-dispersive X-ray spectroscopy (EDS) was performed using a JEOL F200 instrument to observe the surface morphologies and microstructures. The specific surface areas of pristine graphite (AG) and x−MoS_2_@AG composites were obtained by using the Brunauer–Emmett–Teller (BET) method, and pore size distribution was calculated using Barrett–Joyner–Halenda (BJH) method (BELSORP Max, Microtrac MRB, Osaka, Japan). Thermogravimetric analysis (TGA) was carried out on a TGA2 (Mettler Toledo) at a heating rate of 10 °C min^−1^ from 30 to 700 °C under an air atmosphere.

### 2.3. Electrochemical Measurements

The electrochemical properties of the MoS_2_@AG composites were evaluated using a CR2032 coin-type half-cell assembled in an argon-filled glove box. A slurry was prepared by mixing the active material (MoS_2_@AG composites), conducting agent (Super P), and polyvinylidene fluoride (PVDF) binder at a weight ratio of 8:1:1 in a solvent of N-methyl-2-pyrrolidone (NMP). The resulting slurry was coated onto a copper foil and dried in a vacuum at 120 °C for 12 h. The average loading density of the anode electrodes was about 2 mg cm^−2^. The anode was composed of the MoS_2_@AG-composite-coated copper foil, while a lithium metal foil was used as the counter/reference electrode. A polypropylene membrane was used as the separator. The electrolyte solution comprised a 1M LiPF_6_ solution in a mixture of ethylene carbonate (EC)/ethylene methyl carbonate (EMC)/diethyl carbonate (DEC) (=3:4:3, *v*/*v*/*v*) containing 1% vinylene carbonate (VC). The electrochemical cell was fabricated using 160 μL of electrolyte. The galvanostatic charge and discharge curves of the cells were obtained using a battery testing system (Biologic, BCS) in a voltage range from 0.01 to 2.5 V vs. Li/Li^+^ with various current densities. The electrochemical impedance spectroscopy (EIS) measurements were performed at an AC amplitude of 10 mV over the frequency range from 200 kHz to 50 mHz (Biologic, VSP) and the equivalent circuit fitting was conducted using ZView Software 3.2 (Scribner Associates, Inc., Southern Pines, NC, USA).

## 3. Results and Discussion

Figure 1a illustrates the one-step hydrothermal synthesis process for forming 3D hierarchical flower-like MoS_2_ nanosheets directly grown on the surface of graphite. Figure 1b–d show representative SEM, TEM, and high-resolution TEM (HR-TEM) images of the MoS_2_@AG composites, respectively. The x−MoS_2_@AG composites with different weight percentages of sulfur and molybdenum sources relative to graphite were synthesized. In this process, MoS_2_ nanosheets were spontaneously nucleated and grown on the surface of graphite. Ultimately, a 3D composite of few-layer MoS_2_ nanosheets with a low loading on graphite was obtained. The unique structure of the MoS_2_-decorated graphite composite is anticipated to significantly enhance the capacity and rate capability of LIBs. Graphite acts as a support for MoS_2_ nucleation, thereby providing an excellent electron transfer channel. Additionally, 3D hierarchically aligned flower-like MoS_2_ nanosheets grown on the surface of graphite improve the rate characteristics by facilitating lithium diffusion and increase the specific capacity by offering abundant exposed active sites.

The morphologies of pristine graphite and MoS_2_-decorated graphite composites were examined by SEM and SEM-backscattered electron (SEM-BSE) images to investigate the effect of the amounts of thiourea and (NH_4_)_6_Mo_7_O_24_⋅4H_2_O on the formation process (Figure 2 and Appendix A). Three-dimensionally grown plate-like MoS_2_ nanosheets were observed on the surface of 2.5−MoS_2_@AG and 5−MoS_2_@AG composites, as shown in Figure 2a–c and Figure 2d–f, respectively. In contrast, flower-shaped MoS_2_ nanosheets were observed on the surfaces of 10−MoS_2_@AG and 20−MoS_2_@AG composites (Figure 2g–l). The increase in the amounts of thiourea and (NH_4_)_6_Mo_7_O_24_⋅4H_2_O provided more nucleation sites, resulting in the formation of more flower-like MoS_2_ nanosheets. Appendix A shows the enlarged SEM images of the flower-like MoS_2_ nanosheets. Furthermore, Figure 2c,f,i,l show the SEM-BSE images corresponding to Figure 2b,e,h,k, respectively. The brighter areas indicate higher average atomic numbers, confirming the successful formation of MoS_2_ on the graphite surface.

The specific surface area and pore structure of pristine graphite (AG) and 20−MoS_2_@AG composite were examined by the N_2_ adsorption and desorption isotherm measurements (Appendix A). The Brunauer–Emmett–Teller (BET) specific surface area of the 20−MoS_2_@AG composite was calculated to be 2.98 m^2^ g^−1^, which is almost 2.6 times larger than that of pristine AG (1.15 m^2^ g^−1^). As shown in the inset of Appendix A, the mean pore diameters of AG and 20−MoS_2_@AG were 29.28 and 21.48 nm, respectively, and 20−MoS_2_@AG showed narrower average pores. The increased specific surface area and decreased pore diameter of the 20−MoS_2_@AG composite can be attributed to the formation of 3D hierarchically aligned MoS_2_ nanosheets on the surface of graphite. These structural features are expected to improve the rate characteristic by facilitating lithium diffusion and increase the specific capacity by offering abundant exposed active sites.

The amount of MoS_2_ grown on the graphite surface was calculated through thermogravimetric analysis (TGA), as shown in Appendix A. In the case of the pristine MoS_2_, the weight loss from 300 to 500 °C is indicative of the oxidation of MoS_2_ to MoO_3_. As for the 20−MoS_2_@AG composites, the TGA profiles display a two-step weight decrease, which can be attributed to the consecutive oxidations of MoS_2_ and carbon, respectively. The MoS_2_ content in the 20−MoS_2_@AG was estimated to be <5 wt% through the TGA results.

The low-magnification TEM image (Figure 3a) reveals the MoS_2_ nanosheets directly grown on the graphite surface. High-resolution TEM (HR-TEM) images in Figure 3b,c show that the few-layered MoS_2_ nanosheets are grown on the surface of graphite. The MoS_2_ nanosheets consist of 9–14 layers, and the interlayer distance (002) of MoS_2_ was found to be 0.64~0.67 nm, which is slightly larger than that of conventional bulk MoS_2_ (0.62 nm). This expanded interlayer spacing could contribute to enhanced kinetics and a low energy barrier for ion intercalation [25,26,27]. Furthermore, high-angle annular dark-field scanning transmission electron microscopy (HAADF-STEM) images of MoS_2_-decorated graphite are shown in Figure 3d and energy dispersive spectroscopic (EDS) mapping images of MoS_2_-decorated graphite reveal the existence of C, Mo, and S elements in the composite, as shown in Figure 3e–g.

Figure 4a shows the powder XRD pattern of AG, 2.5−MoS_2_@AG, 5−MoS_2_@AG, 10−MoS_2_@AG, and 20−MoS_2_@AG. The diffraction peaks observed in all MoS_2_-decorated graphite composites at 26.44°, 42.30°, 44.48°, and 54.58° were consistent with the (002), (100), (101), and (004) planes of hexagonal graphite (JCPDS 41-1487). Additionally, the (002) diffraction peak shifted to a lower angle compared to that of pristine graphite (26.58°), indicating an increased interlayer spacing (Appendix A). The three-dimensionally aligned few-layer MoS_2_ nanosheets grown on the surface of graphite are expected to provide more space for Li^+^ ion transport and reduce the kinetic barriers for their movement. Figure 4b shows an enlarged XRD pattern of the low-angle region in Figure 4a, and the peaks at 14.22° and 33.24° match well with the (002) and (101) planes of hexagonal MoS_2_ (JCPDS 37-1492). The average interlayer spacing (002) of MoS_2_ calculated from Bragg’s law is about 0.622 nm, which is slightly longer than that of highly crystalline MoS_2_ (0.615 nm).

The Raman spectra of MoS_2_-nanosheets-decorated graphite are presented in Figure 4c, showing two sharp peaks at 380 and 407 cm^−1^ attributed to the E^1^_2g_ and A_1g_ vibration modes of MoS_2_, respectively [28]. Two characteristic bands were also observed at 1358 and 1580 cm^−1^, corresponding to the D band and G band of graphite, respectively.

The XPS measurements were used to analyze the elemental composition of MoS_2_-decorated graphite. The XPS survey spectra in Figure 4d show that MoS_2_-decorated graphite contains Mo, S, C, and O elements. In the high-resolution Mo 3d spectrum (Figure 4e), two peaks are observed at 229.7 and 232.8 eV, which can be attributed to the Mo 3d_5/2_ and Mo 3d_3/2_ binding energies, respectively, and are characteristic peaks of Mo^4+^ in MoS_2_. The peaks at 232.9 and 236.1 eV are related to the Mo 3d_5/2_ and 3d_3/2_ of Mo^6+^ (typical of the Mo-O bond). Furthermore, a small S 2s peak is displayed at 226.7 eV. The high-resolution S 2p spectrum is consistent with S 2p_1/2_ at 163.6 eV and S 2p_3/2_ at 162.5 eV, respectively (Figure 4f) [22].

The electrochemical performance of the MoS_2_-decorated graphite composite was evaluated as an LIB anode material. Figure 5a shows the galvanostatic charge and discharge (GCD) profiles of the 20−MoS_2_@AG electrode during the initial first cycle in a voltage range of 0.01–2.5 V vs. Li/Li^+^ at a current density of 35 mA g^−1^. The 20−MoS_2_@AG electrode exhibited an initial charge capacity of 494.1 mAh g^−1^ with a coulombic efficiency of 86.1%. The capacity of 20−MoS_2_@AG is higher than that of pristine graphite (373.3 mAh g^−1^) due to the formation of MoS_2_ nanosheets on the graphite surface. However, the initial coulombic efficiency is slightly lower than that of graphite (~91.2%), indicating that electrochemically active MoS_2_ nanosheets contribute to the total capacity of the 20−MoS_2_@AG composite [12,14]. 

Figure 5b shows the differential voltage profile of the 20−MoS_2_@AG. In the first cycle, the cathodic peaks at 0.08 V and 0.19 V in the discharge (lithiation) process are attributed to the gradual intercalation of Li^+^ into the interlayers of graphite [15,29]. Additional discharge contributions corresponding to the MoS_2_ nanosheets on the graphite surface were observed at higher voltages of ~0.68 and 1.18 V vs. Li/Li^+^. The peak at 1.18 V is ascribed to Li^+^ insertion into the structure of MoS_2_ to form Li_X_MoS_2_. Another peak at 0.68 V corresponds to the reduction/decomposition of MoS_2_ to Mo metal particles and Li_2_S through a conversion reaction (MoS_2_ + 4Li^+^ → Mo + 2Li_2_S), and the formation of the solid electrolyte interphase (SEI) layers [25,30,31]. MoS_2_ reacts with Li^+^ ions at a higher operating voltage compared to graphite. Therefore, the lithiated 20−MoS_2_@AG electrode can lower the Li^+^ adsorption energy on the surface of graphite and form a stable interface between MoS_2_ and graphite, promoting Li^+^ migration during cycling [12,14,29]. Additionally, the slightly higher anodic potential in the MoS_2_@AG composite would prevent the formation of lithium dendrites during repeated cycling, which typically occurs in conventional graphite anodes [29]. In the subsequent reverse anodic scan, two peaks at 1.63 V and 2.23 V are associated with the incomplete oxidation of Mo metal into MoS_2_ and the partial de-lithiation of Li_2_S into S, respectively. In the next cycle, the reduction peaks at 0.68 V and 1.18 V are disappeared and two new peaks at 1.1 V and 1.95 V are observed, assigned to the following reactions: 2Li^+^ + S + 2e^−^ → Li_2_S and MoS_2_ + xLi^+^ + xe^−^ → Li_x_MoS_2_, respectively [25,32]. These peak shifts could be explained by the structural change and phase transformation during the first cycling [28]. Even at the 100th cycle, it was confirmed that two weak peaks at 1.3 V and 1.88 V were still observed (Appendix A). As shown in Appendix A, the pristine MoS_2_ electrode shows a similar differential voltage profile to the 20−MoS_2_@AG electrode. This means that the 20−MoS_2_@AG composite electrode has an energy storage mechanism combined with those of graphite and MoS_2_. Moreover, the presence of two main pairs of redox peaks during cycling indicated the stable cycle stability and reversibility of the 20−MoS_2_@AG electrode.

Figure 5c and Appendix A show the cycle stabilities of pristine AG, pristine MoS_2_, 5−MoS_2_@AG, 10−MoS_2_@AG, 20−MoS_2_@AG, and 30−MoS_2_@AG electrodes, with their initial charge (de-lithiation) capacities at a current density of 200 mA g^−1^ being 337.1, 812.0, 407.0, 418.0, 448.0, and 523.2 mAh g^−1^, respectively. The highest specific capacity of the 30-MoS_2_@AG can be ascribed to the greater loading mass of MoS_2_. The pristine flower-like MoS_2_ nanosheets electrode exhibited a rapid capacity decay during initial cycles, likely due to its low conductivity and severe structural damage caused by volume changes, leading to the pulverization of the active material [33]. In contrast, the 20−MoS_2_@AG composite electrode displayed excellent cycle performance with an average coulombic efficiency of 99.3% and delivered the highest reversible capacity of 463.2 mAh g^−1^ after 100 cycles. The reversible capacity of the 20−MoS_2_@AG electrode is much larger than the theoretical capacity of 386.9 mAh g^−1^ (C_20−MoS2@AG_ = C_graphite_
× wt% of graphite (95%) + C_MoS2_
× wt% of MoS_2_ (5%) = 372 × 0.95 + 670 × 0.05 = 386.9 mAh g^−1^). This can be attributed to the formation of 3D hierarchical MoS_2_ nanosheets on the graphite surface, resulting in abundant active sites for Li^+^ diffusion, expanded interlayer spacing, and a large electrode/electrolyte contact area [25,31].

The galvanostatic charge and discharge curves of pristine MoS_2_ and 20−MoS_2_@AG electrodes at current densities of 35 and 200 mA g^−1^ are presented in Appendix A. Interestingly, both pristine MoS_2_ nanosheets and x−MoS_2_@AG composite electrodes exhibited a capacity climbing phenomenon during cycles, which is likely due to the increasing electrochemically accessible surface area, resulting from the gradual appearance of cracks on the (002) basal planes of MoS_2_ [33,34,35,36,37].

The rate capabilities of pristine AG, pristine MoS_2_, 5−MoS_2_@AG, 10−MoS_2_@AG, 20−MoS_2_@AG, and 30−MoS_2_@AG electrodes were evaluated at current densities of 200, 400, 800, 1200, and 2400 mA g^−1^ (1C = ~400 mA g^−1^), as shown in Figure 5d and Appendix A. Among them, the 20−MoS_2_@AG electrode demonstrated the best rate performance compared to pristine AG and pristine MoS_2_. At current densities of 200, 400, 800, 1200, and 2400 mA g^−1^, the average charge capacities of 20−MoS_2_@AG delivered reversible capacities of 441.7, 431.4, 413.1, 371.0, and 255.8 mAh g^−1^, respectively, which were superior to those of pristine MoS_2_ (699.5, 349.2, 198.8, 139.7, and 84.1 mAh g^−1^) and pristine AG (365.5, 342.7, 256.8, 172.6, and 92.5 mAh g^−1^). The 20−MoS_2_@AG electrode displayed a high reversible capacity of 371.0 mAh g^−1^ at a high current density of 1200 mA g^−1^. Notably, the 20−MoS_2_@AG electrode maintained a high-capacity value even under high-rate conditions, retaining approximately 84.0% of the capacity compared to that at a relatively low current density of 200 mA g^−1^ (Appendix A). Even when the current density recovered to 200 mA g^−1^ after 50 cycles, the capacity of the 20−MoS_2_@AG electrode remained at 439.5 mAh g^−1^, indicating excellent reversibility and rate cycle stability. On the other hand, the 30-MoS_2_@AG electrode with a higher amount of MoS_2_ showed more capacity fading as the current density increased (71.0% retention of the capacity at 1200 mA g^−1^). These results showed that the higher the MoS_2_ content, the lower the rate characteristics due to its low intrinsic electronic conductivity and large volume change upon cycling. Therefore, it was determined that the 20−MoS_2_@AG composite was the most optimal condition. Additionally, all MoS_2_-decorated graphite electrodes exhibited enhanced rate performance and reversibility. This can be attributed to the combination of graphite’s excellent conductivity and the flower-like MoS_2_ nanosheets grown on its surface, which increases the contact/accessible area with the electrolyte, provides more reaction sites, and facilitates Li^+^ diffusion, thereby contributing to the excellent electrochemical performance [26].

A long-term cycling performance at a high current density is essential for practical applications of LIBs [25]. Figure 5e illustrates the cycle performance of graphite and 20−MoS_2_@AG electrodes at a high current density of 1200 mA g^−1^ over 300 cycles. The 20−MoS_2_@AG electrode exhibits an initial capacity of ~400 mAh g^−1^, higher than that of AG (~248 mAh g^−1^), and shows stable cycle performance over 300 cycles. The significant improvement in the overall performance of x−MoS_2_@AG composites compared to pristine graphite is mainly due to the surface decoration with MoS_2_ nanosheets. Furthermore, the 20−MoS_2_@AG electrode exhibited stable cycle characteristics over 400 cycles at a high current density of 1200 mA g^−1^, while the 30−MoS_2_@AG electrode showed a rapid decrease in capacity from around the 315th cycle (Appendix A).

The electrochemical performance of our MoS_2_@AG as anode materials of LIBs was compared with those of the previously reported transition metal sulfides (MS_x_) or transition metal oxides (MO_x_)-based carbon composites (Appendix A) [38,39,40,41,42]. Upon comparing the specific capacity of our MoS_2_@AG composite anode to that of carbon composites incorporating transition metal sulfides or transition metal oxides, we observed that the capacity of our composite was relatively lower than carbon-based composites containing more than 50 wt% of transition metal sulfides or oxides. However, in comparison to anode materials that contained small amounts (<10 wt%) of transition metal sulfides or oxides, our composite demonstrated superior capacity characteristics under not only low current density conditions, but also high current density conditions. These findings are expected to provide the MoS_2_@AG composite with a small amount of MoS_2_ surface modification on the graphite surface as a potential anode material for high-performance and cost-effective lithium-ion batteries.

Electrochemical impedance spectroscopy (EIS) was used to obtain further insight into the electrochemical reaction kinetics of AG, pristine MoS_2_ nanosheets, and 20−MoS_2_@AG electrodes (Figure 5f). EIS plots were analyzed based on the fitting results obtained using an equivalent circuit model (inset of Figure 5f). The Nyquist plots of the electrodes can be divided into four components. The intercept at the real axis in the high-frequency region represents the internal resistance (R_s_), a semicircle in the high-frequency region corresponds to the surface film resistance (R_f_), a semicircle in the medium-frequency region represents charge transfer resistance (R_ct_), and a straight line in the low-frequency region represents the Warburg impedance (Z_w_) related to the diffusion resistance of Li^+^ within the bulk of the electrode materials [15,21,29]. CPE represents a constant phase element, corresponding to the charge-transfer reaction. The values for the resistance parameters obtained from fitting using the equivalent circuit are summarized in Appendix A. The diameter of the semicircle (R_f_ + R_ct_) of the 20−MoS_2_@AG electrode (128 Ω) in the high/medium frequency is similar to that of AG (101 Ω). Moreover, the slopes of the straight lines at low frequencies for AG and 20−MoS_2_@AG are larger than those for pristine MoS_2_ nanosheets, indicating that the MoS_2_@AG electrode has a faster diffusion rate at the interface. The equivalent circuit fitting result confirms that both Warburg resistance (Z_w_R (Ω)) and Warburg time constant (Z_w_T (s)) values of AG and MoS_2_@AG electrodes are about 2–5 times smaller than those of MoS_2_. The ion diffusion coefficient is inversely proportional to the inverse ratio values of Z_w_T and Z_w_R^2^ [43]. In addition, the Nyquist plots of the pristine AG, pristine MoS_2_, and 20−MoS_2_@AG electrodes were obtained after two cycles at a current density of 35 mA g^−1^, as shown in Appendix A. The pristine AG and 20−MoS_2_@AG after two cycles show a decrease in both surface film and charge-transfer resistance (R_f_ + R_ct_) showing 101 → 76.3 Ω for AG and 128 → 76.8 Ω for MoS_2_@AG. This observation is attributed to the improved infiltration of electrolyte into electrode materials and charge transfer kinetics. However, the R_f_ + R_ct_ of cycled MoS_2_ electrode increases slightly (128 → 139 Ω), which corresponds mainly to the rapid capacity decay due to the pulverization of active materials.

Examining the changes in surface and thickness of the electrode before and after cycling provides a visual means to understand the mechanism underlying the improved performance of the MoS_2_ nanosheets decorated on the graphite composite in comparison to pristine graphite. In light of this, we investigated the structural stability of the 20−MoS_2_@AG electrode before and after cycling by performing SEM measurements, as shown in Appendix A. Prior to cycling, the MoS_2_@AG electrode had a rough surface covered with numerous nanosheets and its thickness was approximately 36.2 μm. After 300 cycles at a current density of 200 mA g^−1^, the 3D hierarchically aligned MoS_2_ nanosheets synthesized on the surface of the graphite had transformed into nanoparticles and aggregated with each other, but some MoS_2_ nanosheets still maintained their original shape. In addition, we observed that the thickness of the 20−MoS_2_@AG electrode after cycling was about 36.5 μm, which was almost unchanged compared to the initial thickness of 36.2 μm before cycling. These ex situ SEM images provide evidence that the 20−MoS_2_@AG electrode maintains a stable structure without significant volume changes even after long-term cycling, indicating that high reversible capacity can be maintained even after prolonged cycling.

## 4. Conclusions

In conclusion, we have successfully synthesized a 3D hierarchical MoS_2_@AG composite that exhibits high performance as an anode material for LIBs. The use of graphite as a substrate for the nucleation and growth of three-dimensionally aligned few-layer MoS_2_ nanosheets provided a conductive matrix and exposed surface area, resulting in a significant improvement in the rate capability and capacity of the composite. The hydrothermal reaction facilitated the growth of MoS_2_ nanosheets on the graphite surface, forming a 3D network with excellent structural stability during the charge/discharge process. The three-dimensionally grown few-layer MoS_2_ nanosheets on the surface of graphite facilitated the diffusion of Li^+^ and reduced the diffusion resistance, leading to improved rate performance. The 20−MoS_2_@AG electrode exhibited excellent cyclability and rate capability with outstanding stability over 300 cycles, even at a high current density of 1200 mA g^−1^. Notably, at this current density, the 20−MoS_2_@AG composite displayed an initial capacity of around 400 mAh g^−1^, which is approximately 60% higher than that of the graphite electrode. These results demonstrate that the MoS_2_-nanosheets-decorated graphite composite developed through a simple hydrothermal method holds great promise as a high-power anode material for LIBs in various practical applications.

## Figures and Tables

**Figure 1 materials-16-04016-f001:**
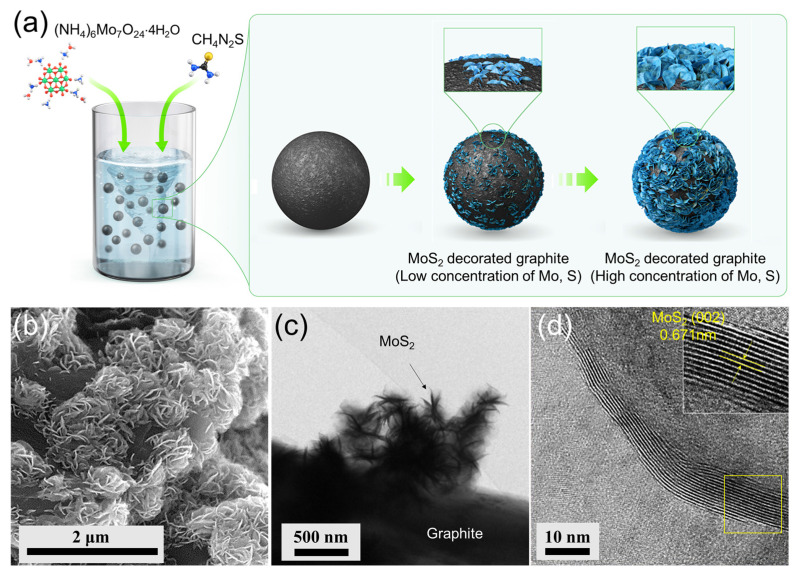
(**a**) Schematic illustration and microstructural analysis of one-step hydrothermal synthesis of the 3D hierarchically aligned flower-like MoS_2_ nanosheets grown directly on the surface of graphite. The MoS_2_@AG composite was formed through spontaneous nucleation and growth of few-layer MoS_2_ nanosheets on the surface of graphite. Representative images of the composite obtained by (**b**) SEM, (**c**) TEM, and (**d**) high-resolution TEM (HR-TEM), revealing the morphology and structure of the composite at different scales.

**Figure 2 materials-16-04016-f002:**
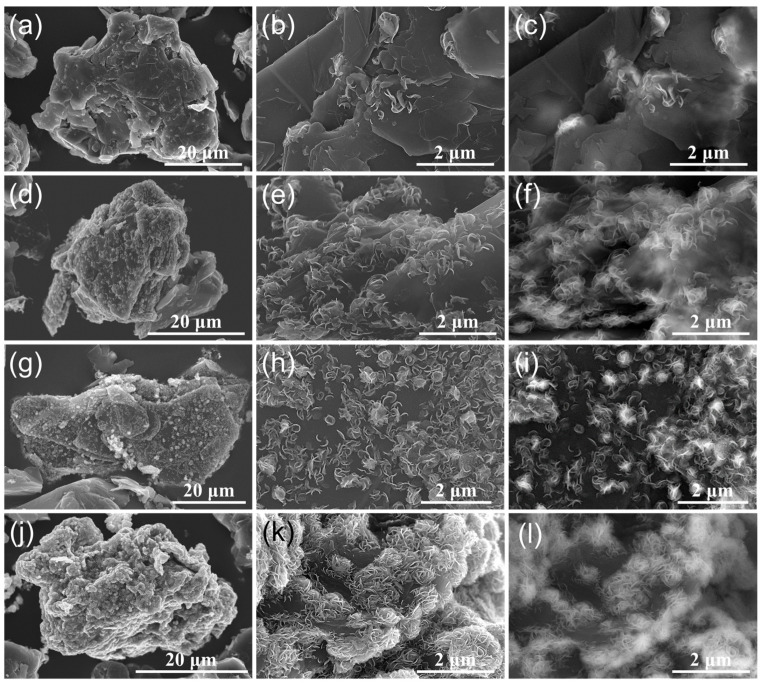
SEM and SEM-BSE images of MoS_2_-nanosheets-decorated graphite composites with varying amounts of thiourea and (NH_4_)_6_Mo_7_O_24_⋅4H_2_O: (**a**–**c**) 2.5−MoS_2_@AG, (**d**–**f**) 5−MoS_2_@AG, (**g**–**i**) 10−MoS_2_@AG, and (**j**–**l**) 20−MoS_2_@AG. The images show vertically aligned plate-like MoS_2_ nanosheets on 2.5−MoS_2_@AG and 5−sMoS_2_@AG composites, and flower-shaped MoS_2_ nanosheets on 10−MoS_2_@AG and 20−MoS_2_@AG composites. The SEM-BSE images in (**c**,**f**,**i**,**l**) confirm the successful formation of MoS_2_ nanosheets on the graphite surface by brighter contrast indicating a higher atomic number.

**Figure 3 materials-16-04016-f003:**
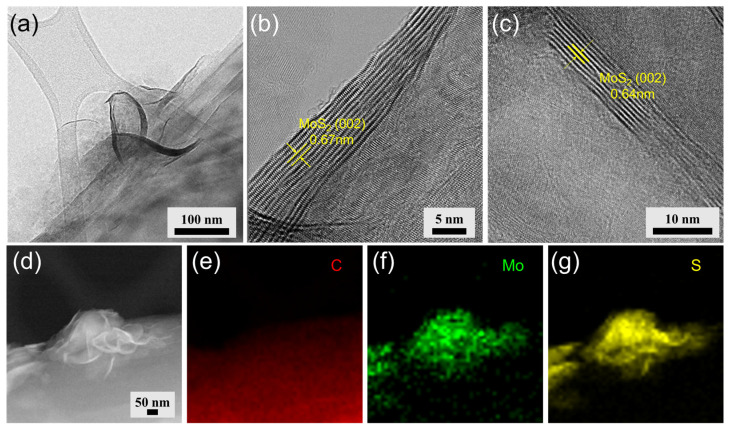
TEM and STEM images of MoS_2_-nanosheets-decorated graphite composites. (**a**) Low-magnification TEM image showing 3D architectural MoS_2_ nanosheets formed on the graphite surface. (**b**,**c**) HR-TEM images displaying few-layered MoS_2_ nanosheets grown on the graphite surface. (**d**) STEM-HAADF image and (**e**–**g**) TEM-EDS mapping images indicating the distribution of C, Mo, and S elements in the composite.

**Figure 4 materials-16-04016-f004:**
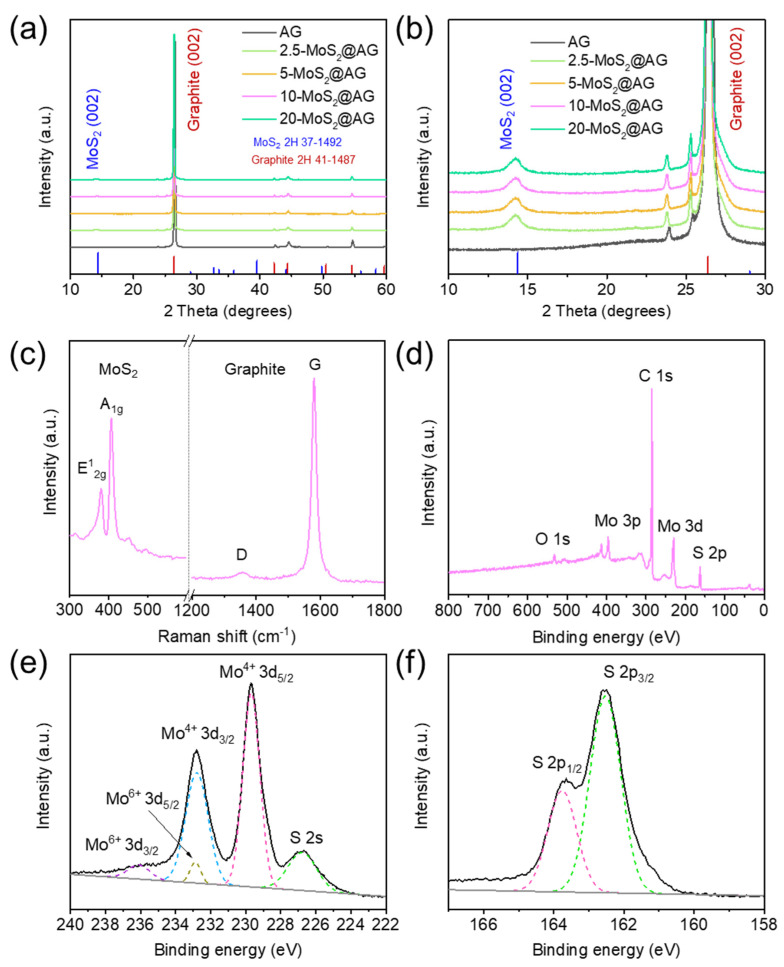
(**a**,**b**) XRD patterns, (**c**) Raman spectra, (**d**) XPS survey spectrum, and high-resolution XPS spectra of (**e**) Mo 3d and (**f**) S 2p for MoS_2_@AG composites.

**Figure 5 materials-16-04016-f005:**
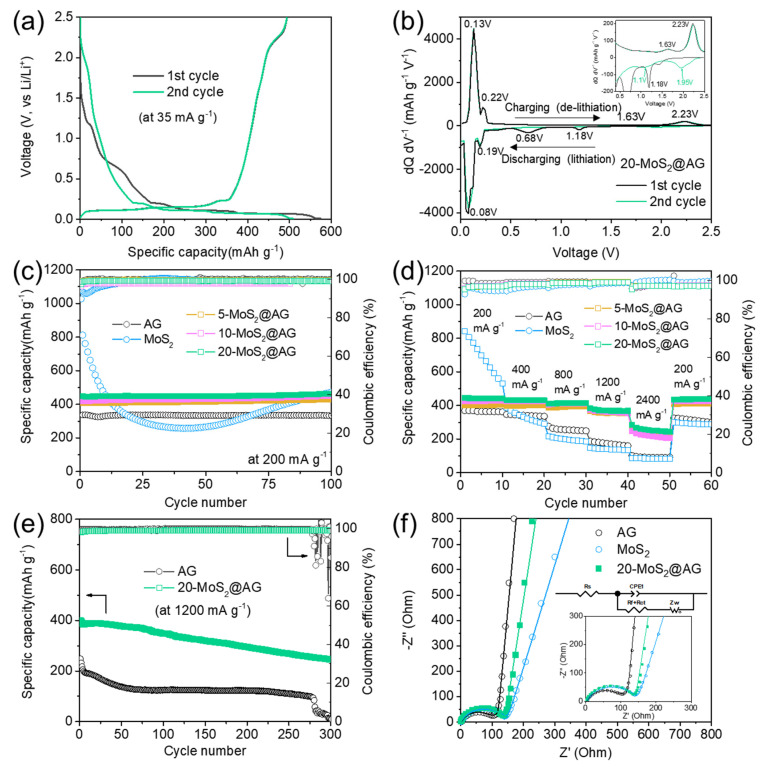
(**a**) Initial galvanostatic charge and discharge curves at a current density of 35 mA g^−1^ and (**b**) corresponding differential voltage profiles of the 20−MoS_2_@AG electrode, (**c**) cycle performance at a current density of 200 mA g^−1^, (**d**) rate performance comparison of pristine AG, pristine MoS_2_, and MoS_2_@AG composite electrodes at different current densities ranging from 200 to 2400 mA g^−1^, (**e**) cycle performance at a high current density of 1200 mA g^−1^ for pristine AG and 20−MoS_2_@AG electrodes, and (**f**) Nyquist plots from EIS data (symbols) and fitting results (solid lines) of pristine AG, pristine MoS_2_, and 20−MoS_2_@AG electrodes before the cycle.

## Data Availability

The data are available upon request from the corresponding authors.

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
