# Peer review of "Three-Dimensional Flower-like MoS2 Nanosheets Grown on Graphite as High-Performance Anode Materials for Fast-Charging Lithium-Ion Batteries"

_materials, 2023, doi:10.3390/ma16114016_

Round 1

Reviewer 1 Report

Yeong A. Lee et al studied 3D flower-like MoS2 nanosheets grown on graphite as high-performance anode materials for fast-charging lithium-ion batteries. The work shows interesting results and can be considered for acceptance after proper revision.

1. According to the current results, with the increase of molybdenum sulfide ratio, the battery capacity and cycle performance improved. Why is the composition only set at 20%? Will a further increase in the MoS2 proportion lead to better electrochemical performance?

2. BET tests should be provided.

3. More experimental details should be given, such as the loading of electrolyte.

4. Fig. 5f. Equivalent circuit diagrams are suggested to provide.

5. The comparison towards the electrochemical performance of related materials should be presented in a table.

6. SEM images of the electrode plate before and after the cycle should be provided.

7. Some references are too old and should be renewed. More metal sulfides need to be mentioned in different energy storage applications.

[1] Microporous and Mesoporous Materials 2021, 316, 110924. https://doi.org/10.1016/j.micromeso.2021.110924

[2] Rare Metals 2022,41(12), 3946–3956.http://doi.org/10.1007/s12598-022-02167-y

[3] Journal of Materials Science & Technology 2023, 154, 1-8. http://doi.org/10.1016/j.jmst.2022.12.042

Minor editing of English language required.

Author Response

May 15, 2023

Ms. Ivona Sindjelic

Special Issue Editor

Materials

RE: Manuscript ID: materials-2383181 “Three-Dimensional Flower-like MoS2 Nanosheets Grown on Graphite as High-Performance Anode Materials for Fast-Charging Lithium-ion Batteries”

Dear. Editor

We are submitting our manuscript entitled “Three-Dimensional Flower-like MoS2 Nanosheets Grown on Graphite as High-Performance Anode Materials for Fast-Charging Lithium-ion Batteries” that has been revised to address all the comments raised by reviewers.

The authors appreciate the reviewer’s valuable time and efforts for going through the manuscript and making constructive comments on it. The manuscript has been carefully revised according to the reviewer’s comments and the point-to-point responses for all the comments are discussed below (Reviewers’ remarks are reproduced in blue while our responses are summarized in red).

We have included a marked copy of the revised manuscript showing the modified and added sections highlighted in yellow as a for-review-only material along with the final revised manuscript and the revised supplementary information. We hope that the revised manuscript successfully addresses all the comments and concerns of the editor and the reviewer.

Thank you for your time and consideration.

Sincerely,

Hana Yoon, Ph. D.

Principal Researcher, Korea Institute of Energy Research (KIER)

Adjunct Professor, Graduate School of Energy Science and Technology (GEST), Chungnam National University

152, Gajeong-ro, Yuseong-gu, Daejeon, 34129, Korea

Tel: +82-42-860-3201

Mobile: +82-10-8767-8239

Kyubock Lee, Ph. D.

Associate Professor, Graduate School of Energy Science & Technology (GEST), Chungnam National University,

99 Daehak-ro, Yuseong-gu, Daejeon 34134, Korea

Tel: +82-42-821-8610

Mobile: +82 10-4488-8460

Hyeyoung Shin, Ph. D.

Assistant Professor, Graduate School of Energy Science & Technology (GEST), Chungnam National University,

99 Daehak-ro, Yuseong-gu, Daejeon 34134, Korea

Tel: +82-42-821-8602

Mobile: +82 10-9337-7355

Reviewer 2 Report

In the manuscript “Three-Dimensional Flower-like MoS2 Nanosheets Grown on Graphite as High-Performance Anode Materials for Fast-Charging Lithium-ion Batteries”, Lee et al. display the synthesis of MoS2@graphite as a new anode material for LIB. The main advantages of this material, claimed by the authors, are the rate capability and cycling stability. However, some issues arise in this work need to be solved before the manuscript can be accepted:

1.     MoS2 was already shown previously as an anode material for LIB. Yet the authors don’t mention or doing any comparison to previous work to justify the superior performance of their work.

2.     it was observed that the mass loading of MoS2 on graphite remains low, i.e., less than 16%, implying that the majority of the capacity originates from the graphite. Therefore, enhanced performance relative to pristine graphite and MoS2 is attributed to other mechanisms potentially induced by MoS2, rather than the MoS2 intrinsic capacity. The authors are urged to provide an explanation for this phenomenon. Otherwise, the justification for the inclusion of such a low loading of active material, namely MoS2 in this case, in the electrode design is not a good strategy.

3.     The capacity of pristine MoS2 seems to increase and may surpass the capacity of the MoS2@AG anode. This makes the proposed material design questionable.

4.     The RCT of AG after 2 cycles, calculated from the impedance, is better than that of the MoS2@AG anode. The authors should explain why.

5.     The authors have extensively characterized the synthesized material, however, aside from the electrochemical assessment of its performance, no analysis was conducted on the anode post-cycling to validate the charge storage mechanism.

Author Response

(The authors gave the same response as above.)

Reviewer 3 Report

MoS2 has been explored as the active material for various devices including energy storage due to its excellent structural, electronic, and electrochemical features. The layered structure and weak van der Waals interactions can enable the lithium ions to easily access the interlayer spacing of MoS2, leading to a fast redox process without a significant volume change. However, such as low conductivity and inherent stacking of MoS2 sheets limits its applications. To resolve these issues, various carbonaceous materials, including carbon nanotubes, activated carbon and graphene, have been employed to support the MoS2 nanostructures. In the submitted work, the authors (of more than dozen 13 authors !!!) examined flower-like MOS2 nanosheets grown on the surface of graphite anode. Although the reported molybdenum disulfide synthesized hydrothermally on graphite anode are excellent candidates for lithium-ion batteries and cost-effective but throughout the manuscript the novelty part and how the current work advances to the well-known MoS2 reported in the literature is lacking.

This is a serious concern. Actually, heaps of publications are available in the domain for MoS2 derived on graphite anodes (for anstance doi.org/10.3390/molecules28062775; doi.org/10.1002/aenm.201702383).

Therefore, the demonstrated MoS2 material is a significant area of research but less novel unless otherwise, the authors emphasize this in the sections like abstract and introduction. In the results and discussion, provide a table in which different molybdenum disulfides and their performances must be compared and benchmarked before considering further. The manuscript has contained some language issues, please check.

The following specific points need to be considered.

·         The theoretical capacity of this material must be given and how much reversible capacity has been obtained?

·         In the section abstract, line 28 – why it is called “artificial” graphite?

·         The electrochemical reactions involved in the lithium-ion storage performance must be given.

·         After repeated cycles, do the authors observe the decomposition of LixMoS2 into metallic Mo nanoparticles and amorphous Li2S?

·         What is the role of having a MoS2/graphite composite?

·         Are the SEM images representing "flower"-like morphology that appear to be urchin-type? Please check.

·         The observed electrochemical performance can be compared to that of the molybdenum used in binary transition metal oxides as anodes for battery systems such as (doi.org/10.1021/acsami.0c13755).

·         In Figure 5, As observed in a range of other dichalcogenide anode materials, whether the peaks associated with the conversion reactions disappeared after the first cycle? Please explain.

·         Why the observed capacity is lower than that of the reported values for this material in the domain?

Some edits required.

Author Response

(The authors gave the same response as above.)

Round 2

Reviewer 1 Report

The authors have revised the paper as suggested by the reviewers, in which case the paper can be considered for acceptance now.

Reviewer 2 Report

The authors addressed my concerns. 

Please recheck the English before the final submission.

Reviewer 3 Report

The revised version is suitable for publication.